# MicroRNA-9 controls dendritic development by targeting REST

**Sebastian A Giusti[1], Annette M Vogl[1], Marisa M Brockmann[1,2], Claudia A Vercelli[3], Martin L Rein[4], Dietrich Trümbach[5], Wolfgang Wurst[5,6,7,8], Demian Cazalla[9], Valentin Stein[2], Jan M Deussing[4], Damian Refojo[1]\***

[1]Department of Molecular Neurobiology, Max Planck Institute of Psychiatry, Munich, Germany; [2]Institute of Physiology, University of Bonn, Bonn, Germany; [3]Department of Molecular Neurobiology, Instituto de Investigación en Biomedicina de Buenos Aires (IBioBA)-CONICET-Partner Institute of the Max Planck Society, Buenos Aires, Argentina; [4]Department of Neurobiology of Stress and Neurogenetics, Max Planck Institute of Psychiatry, Munich, Germany; [5]Institute of Developmental Genetics, Helmholtz Zentrum München, Neuherberg, Germany; [6]Technische Universität, Munich, Germany; [7]Deutsches Zentrum für Neurodegenerative Erkrankungen, Munich, Germany; [8]Munich Cluster for System Neurology, Munich, Germany; [9]Department of Biochemistry, University of Utah School of Medicine, Salt Lake City, United States

**Abstract** MicroRNAs (miRNAs) are conserved noncoding RNAs that function as posttranscriptional regulators of gene expression. miR-9 is one of the most abundant miRNAs in the brain. Although the function of miR-9 has been well characterized in neural progenitors, its role in dendritic and synaptic development remains largely unknown. In order to target miR-9 in vivo, we developed a transgenic miRNA sponge mouse line allowing conditional inactivation of the miR-9 family in a spatio-temporal-controlled manner. Using this novel approach, we found that miR-9 controls dendritic growth and synaptic transmission in vivo. Furthermore, we demonstrate that miR-9-mediated downregulation of the transcriptional repressor REST is essential for proper dendritic growth.

**\*For correspondence:** refojo@mpipsykl.mpg.de

**Competing interests:** The authors declare that no competing interests exist.

## Introduction

Early neuronal development consists of a stereotypic progression of events involving neurite extension, axonal polarization and growth, dendritic arborization, and synaptic formation (*Urbanska et al., 2008*; *Barnes and Polleux, 2009*; *Caceres et al., 2012*). Dendrites are the main site of information input into neurons. The architecture of the dendritic tree and the targeting of dendrites into appropriate territories critically shape the input properties of neurons and are therefore of profound importance for the establishment of neuronal networks (*London and Hausser, 2005*; *Mumm et al., 2006*; *Parrish et al., 2007*; *Spruston, 2008*). The most relevant synaptic specialization of dendrites is dendritic spines. These actin-rich, structurally plastic protrusions represent the major sites of excitatory synaptic contact (*Bonhoeffer and Yuste, 2002*; *Hering and Sheng, 2001*). The development of both, dendrites and spines, represents a multistep process influenced by external signals and intrinsic genetic programs (*Jan and Jan, 2003*; *Wong and Ghosh, 2002*). Emerging data suggest that specific miRNAs exert critical regulatory functions during neuronal differentiation by controlling developmental gene expression programs driving dendritic and spine maturation (*Schratt et al., 2006*; *Fiore et al., 2009*; *Parrish et al., 2009*; *Magill et al., 2010*).

MicroRNAs (miRNAs) have arisen as a powerful class of conserved noncoding RNAs regulating gene expression. They are particularly enriched in the nervous system where they have been shown to influence neuronal development and function (*Giraldez et al., 2005*; *Schratt et al., 2006*; *Han et al., 2013*).

**eLife digest** Messages are sent back and forth in our brains by cells called neurons that connect to each other in complex networks. Neurons develop from stem cells in a complicated process that involves a number of different stages. In one of the final stages, tree-like structures called dendrites emerge from the neurons and connect with neighboring neurons via special junctions called synapses.

A group of small RNA molecules called microRNAs have roles in controlling the development of neurons. One microRNA, called miR-9, is abundant in the brain and is known to be involved in the early stages of neuron development. However, its role in the formation of dendrites and synapses remains unclear.

Giusti et al. studied this microRNA in mice. A length of DNA, coding for an RNA molecule that binds to miR-9 molecules and stops them performing their normal function, was inserted into the mice. These experiments showed that miR-9 is involved in controlling dendrite growth and synaptic function.

To enable a neuron to produce dendrites, miR-9 binds to and interferes with the RNA molecules that are needed to make a protein called REST. This protein is a transcription factor that switches off the expression of other genes so, in effect, miR-9 allows a set of genes that are needed for dendrite growth to be switched on.

The methodology developed by Giusti et al. could be used to study the functions of other microRNAs.

Importantly, miR-9 is one of the most abundant miRNAs in the developing and adult brain (*Lagos-Quintana et al., 2002*; *Krichevsky et al., 2003*). In mammals, miR-9 precursors are encoded by three genes, *miR-9-1*, *miR-9-2,* and *miR-9-3*, located in different chromosomes. After processing, these paralogous transcripts give rise to a unique mature miR-9. Although the function of miR-9 is well established in neural precursors (*Leucht et al., 2008*; *Zhao et al., 2009*; *Delaloy et al., 2010*; *Bonev et al., 2011*; *Coolen et al., 2012*), its role in differentiated neurons is largely unknown. This is surprising since miR-9 levels are higher in mature neurons than in neural precursors (*Liu et al., 2012*), suggesting that important biological processes beyond proliferation or specification are regulated by miR-9. Recently, a role for miR-9 in axonal extension and branching has been described (*Dajas-Bailador et al., 2012*), indicating the importance of miR-9 in the establishment of neuronal networks. However, the involvement of this microRNA in the later stages of neuronal development remains unexplored.

In this study, we aimed to characterize a potential role for miR-9 in dendritic and spine development in vivo. In order to efficiently knock-down miR-9 in a living animal, we employed miRNA sponges as a long-lasting miRNA inhibition strategy. miRNA sponges are transcripts that carry several in tandem copies of a sequence complementary to the miRNA of interest in their 3'UTRs (*Ebert et al., 2007*). When expressed at high levels, miRNA sponges act as specific competitive inhibitors by binding and sequestering the miRNA of interest. In addition, miRNA sponges can be used to monitor miRNA activity when expressed together in the same transcript with a fluorescent protein gene, which is downregulated upon the binding of the miRNA (*Ebert et al., 2007*).

Analysis of the genomic organization or miRNA genes, revealed that approximately 50% of all mammalian miRNAs reside within either protein coding genes or long noncoding RNAs (lncRNAs) genes (*Lagos-Quintana et al., 2001*; *Rodriguez et al., 2004*). In addition, a large proportion of microRNA genes from different families are clustered and expressed as polycistronic transcripts (*Olena and Patton, 2010*). Therefore, it is a major challenge to develop KO models for miRNA genes, since the targeted deletion of a miRNA might affect overlapping protein-coding or lncRNA genes, or alter the expression of neighbor miRNAs, clustered in the same genomic region. Furthermore, many miR-NAs have seed family members encoded at multiple distant loci, which requires that multiple genes have to be simultaneously knocked-out to obtain full ablation and avoid gene compensations based on functional redundancy. These obstacles severely limit the use of conditional targeting strategies, usually based on the Cre-*loxP* system, for the study of miRNA function.

Strategies based on the use of miRNA sponges as dominant negative genetic tools, can overcome the limitations presented by miRNA-gene-KO approaches. Sponges act in *trans* without interfering with the expression of other genes that could localize with the gene that expresses the miRNA of

interest. Moreover, a single sponge transgene allows the simultaneous silencing of 'seed' sequence redundant microRNAs encoded at distant loci.

Here, we describe for the first time a conditional transgenic sponge mouse model. We derived two different strains for in vivo analysis: (i) a reporter mouse line to monitor miR-9 activity with single-cell resolution and (ii) a line for miR-9 loss-of-function studies. Using the latter, we revealed a novel developmental role of miR-9 in the postmitotic differentiation of neurons, as a key regulator of dendritic growth and synaptic transmission in vivo. Furthermore, we show that regulation of the transcriptional repressor REST by miR-9 is essential for normal dendritic maturation.

## Results

### Fully complementary and bulged miR-9 sponges are equally effective

Depending on the configuration of the binding sites, two sponge designs have been described: fully complementary (FC) sponges, which function by both sequestering miRNA and promoting miRNA degradation (*Ameres et al., 2010*); and bulged (Bg) sponge variants, which carry mismatches that prevent base-pairing with the miRNA at positions 9–12, thereby avoiding AGO2-mediated decay of the sponge (*Ebert et al., 2007*) (*Figure 1—figure supplement 1*).

Before studying the role of miR-9 on dendritic and spine development in vivo, we tested the efficiency of GFP-FC and GFP-Bg miR-9 sponge variants as decoy targets and miR-9 reporters in cell culture systems. Both the sponge variants displayed similar efficacies to inhibit miR-9 activity (*Figure 1—figure supplement 2*). We selected FC sponges for further studies since they displayed a higher predicted binding affinity for miR-9 (FC sponge: −42.9 kcal/mol vs Bg sponge: −32 kcal/mol) and a higher sensitivity as reporters (*Figure 1—figure supplement 2*). This last feature might be crucial for precise analysis of miR-9 expression and activity in vivo.

### MiR-9 regulates dendritic development in primary neuronal cultures

To define a potential role of miR-9 in the maturation of dendritic arbors and spines, we used transiently transfected sponge constructs in developing mouse primary hippocampal neurons. Neurons transfected with GFP-FC-miR-9 sponge showed significantly reduced total dendritic length and complexity after 10 days in vitro (DIV) compared to neurons transfected with GFP (spongeless) or a GFP-scrambled control sponge (*Figure 1A–C*). The specificity of the miR-9 sponge was further confirmed, as miR-9 overexpression rescued the phenotype induced by the GFP-FC-miR-9 sponge (*Figure 1A–C*). miR-9*, encoded by the same miRNA precursor as miR-9, has been postulated to contribute to activity-dependent dendritic growth, although no direct evidence was shown (*Yoo et al., 2009*). Interestingly, expression of FC sponges directed to miR-9* showed no effects on dendritic growth (*Figure 1A–C*). In addition, we found that miR-9 overexpression in hippocampal neurons did not affect total dendritic length and had only a mild impact on dendritic complexity (data not shown). Together, these results indicate that miR-9, but not miR-9*, participates in the developmental programs controlling dendritic growth and that endogenous mir-9 levels have a saturating effect of on dendritic maturation.

Further, we evaluated the density and morphology of spines in transfected neurons at DIV21. No effects of miR-9 inhibition on spine development were observed (GFP-scrambled sponge: 6.48 ± 0.64 vs GFP-FC-miR-9-sponge: 5.85 ± 0.48 spines per 10 μm of dendritic segment, n.s., Student's *t* test).

Dendrite differentiation during brain development involves not only the growth and maturation of the dendritic arbors, but also the establishment of a refined specific connectivity pattern with thousands of inputs arriving from presynaptic neurons. In vivo, these processes are under the strong influence of extrinsic factors such as adhesion molecules, proteins of the extracellular matrix, growth factors, glial contacts, and guidance cues (*Jan and Jan, 2003*; *Sudhof, 2008*). These factors are mostly absent in cell culture systems and consequently not modeled in primary neurons. Therefore, to study the effects of miR-9 in vivo, we generated conditional transgenic GFP-FC-miR-9 sponge mice by homologous recombination in the *Rosa26* (*R26*) locus (miR-9-Sp^fl-Stop) (*Figure 2—figure supplement 1*).

### Characterization of miR-9 activity in the adult mouse brain

The miR-9 sponge transgene allowed us to characterize miR-9 activity in the adult mouse brain. The reporter function relies on the miR-9-induced GFP downregulation of the sponge transcript. However, an often underestimated problem of sponge-based reporters is the impossibility to distinguish between a truly downregulated signal reporting miR-9 activity and a non-specific lack of sponge expression due to promoter a inactivity or locus silencing. We therefore developed a reporter mouse line

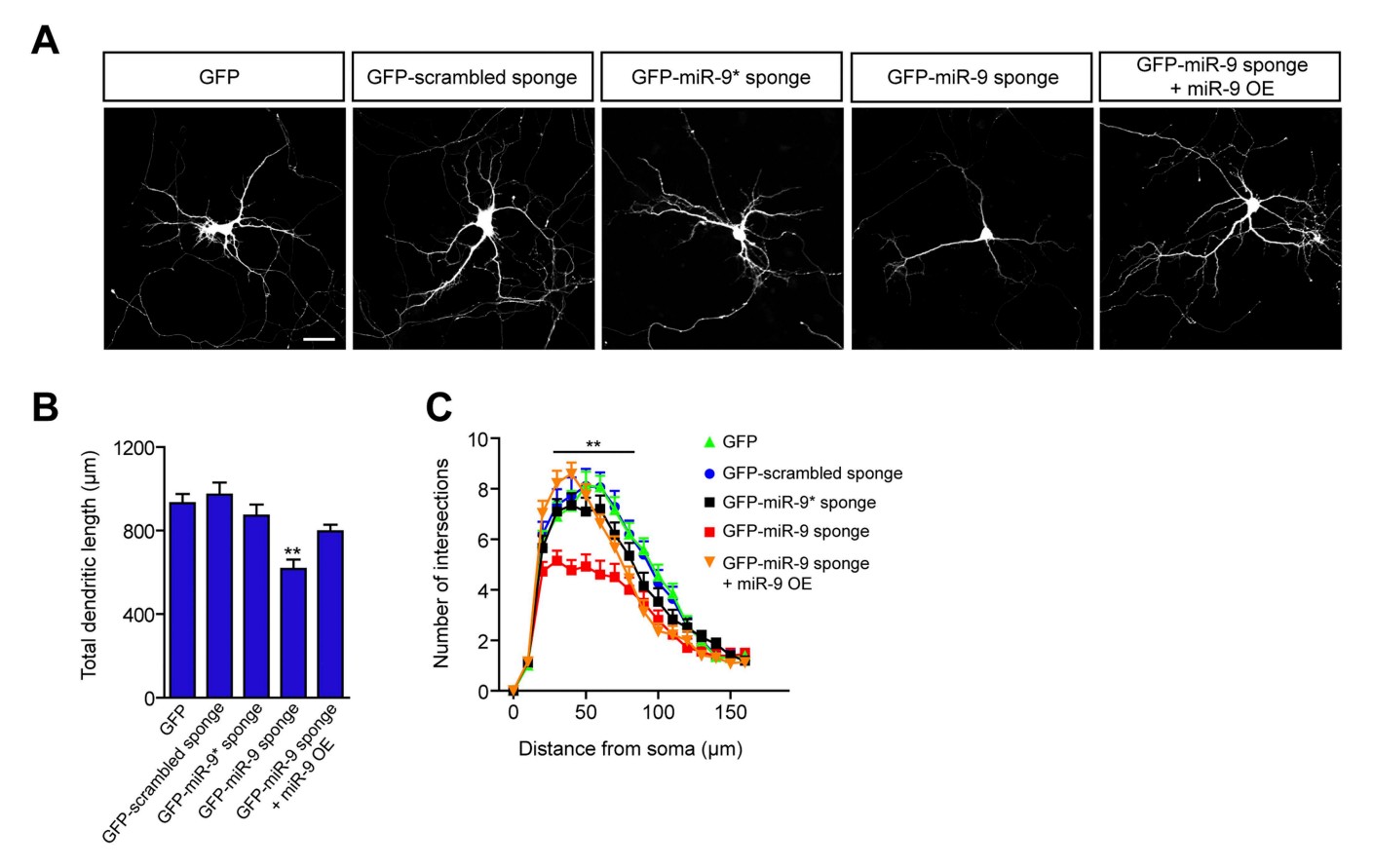

**Figure 1**. miR-9 regulates dendritic development. (**A**) Representative images of primary hippocampal neurons transiently transfected at DIV4 with the indicated constructs and fixed at DIV10. mRFP was cotransfected for visualization. Scale bar 50 μm. (**B**) Quantification of the total dendritic length (mean + SEM, one-way ANOVA, and Bonferroni post-test) and (**C**) quantification of dendritic complexity by Sholl analysis of neurons transfected as described in (**A**) (mean + SEM, repeated measures two-way ANOVA and Bonferroni post-test showing GFP-scrambled vs GFP-miR-9 sponge). **$p < 0.01$, n ≥ 30 neurons per condition.

The following figure supplements are available for figure 1:

**Figure supplement 1**. FC and Bg miR-9 sponges.

**Figure supplement 2**. Comparison of the efficacy of FC and Bg miR-9 sponges.

**Figure supplement 3**. FC and BG sponges as reporters of miR-9 activity.

encoding a fluorescent tdTomato control protein in the other allele of the *R26* locus (see details in *Figure 2—figure supplement 2*). The tdTomato transgene acts as a control allele since it is located in the same locus and is also driven by a CAG promoter. The absence of GFP signal would therefore imply miR-9 activity only if the tdTomato signal is present, excluding lack of expression of the sponge transgene. Histological evaluation of the miR-9-Sp/tdTomato reporter mouse line revealed a strong signal for both proteins in most organs, indicating the absence of miR-9 activity (*Figure 2A*). Lack of expression of both transgenes was observed in seminiferous tubules and in follicular cells of the ovary due to either silencing of the *R26* locus or inactivity of the CAG promoter. Interestingly, we observed a scattered population of GFP-negative/tdTomato-positive hepatocytes, suggesting that a subpopulation of liver cells express miR-9.

A robust downregulation of GFP expression was observed throughout the entire brain, indicating elevated miR-9 activity (*Figure 2B,C*). In particular, GFP expression was strongly downregulated in

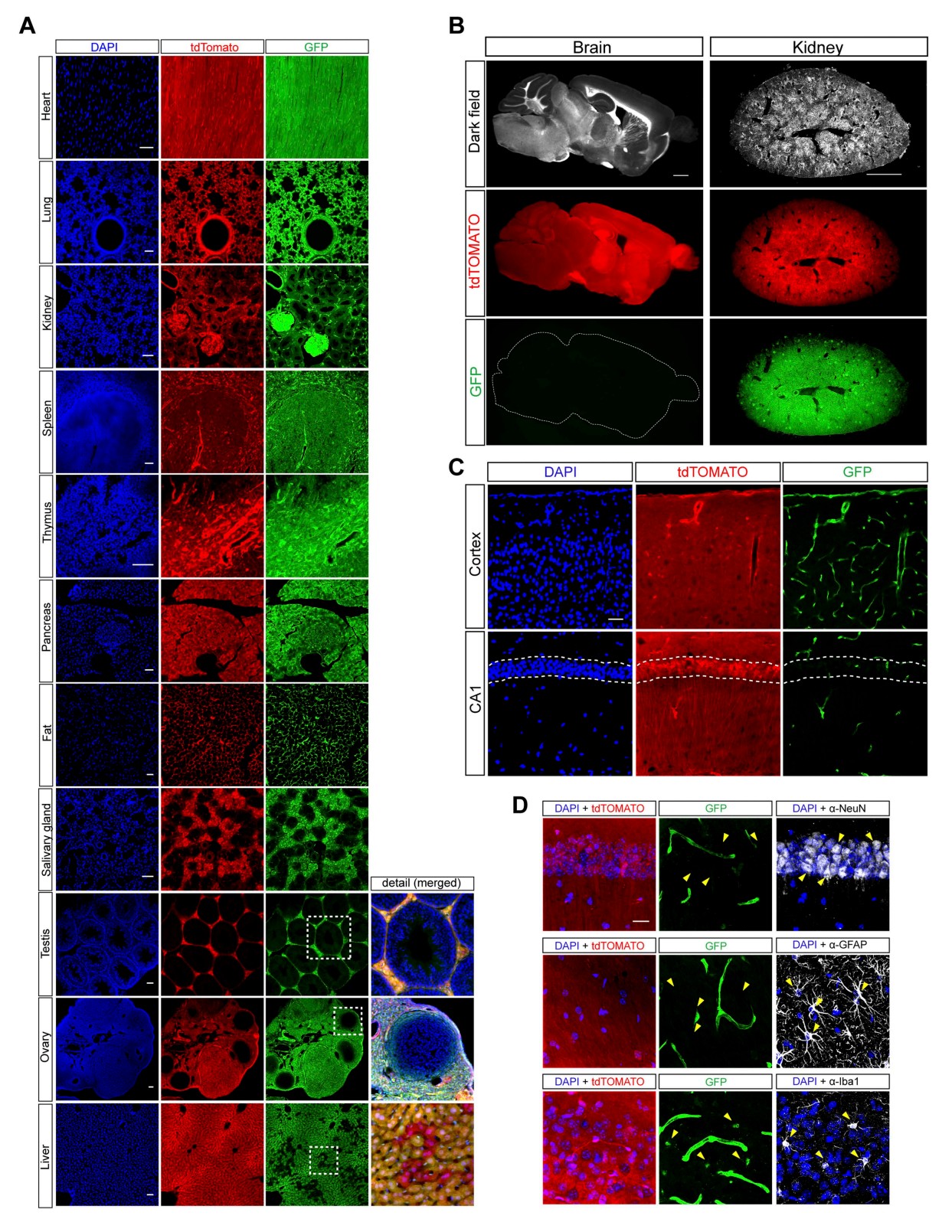

**Figure 2**. Histological analysis of miR-9-Sp/tdTomato reporter mouse line. (**A**) Confocal images of different organs of the reporter mice showing native GFP and tdTomato fluorescence. Scale bar: 50 µm. (**B**) Low power images of sections from brain and kidney showing native GFP and tdTomato fluorescence. Scale bar: 1 mm. (**C**) At microscopic level within the brain, strong native GFP signal could be only detected in blood vessels (top) and in the
*Figure 2. Continued on next page*

*Figure 2. Continued*

choroid plexus. Note the downregulation of GFP in the *stratum pyramidale* (dashed lines) of hippocampal CA1 (bottom). Scale bar: 50 μm. (**D**) Brain sections showing native GFP and tdTomato fluorescence with immunohistochemistry for different cell-specific markers: NeuN (neurons), GFAP (astrocytes), and Iba1 (microglia). Arrows show examples of individual cells. Note the faint native GFP signal in microglial cells. Scale bar: 20 μm.

The following figure supplements are available for figure 2:

**Figure supplement 1**. Generation of conditional GFP-FC-miR-9 sponge mice.

**Figure supplement 2**. miR-9 reporter mouse line.

neurons and astrocytes while we observed a faint GFP signal in microglia (*Figure 2D*). These results suggest that miR-9 expression levels are higher in neurons and astrocytes than in microglia.

## miR-9 sponge transgenic mice exhibit dendritic growth defects and impaired synaptic transmission

To address the impact of miR-9 loss of function on dendritic growth in vivo, we generated a miR-9-Sp$^{Nestin-Cre}$ mouse line by breeding miR-9-Sp$^{fl-Stop}$ mice with the Nestin-Cre mouse line. This newly generated mouse line conditionally expresses the GFP-FC-miR-9-sponge in the CNS. Unlike the reporter line, miR-9-Sp$^{Nestin-Cre}$ mice are homozygous for the sponge transgene and lack a tdTomato allele. miR-9-Sp$^{Nestin-Cre}$ mice are viable and fertile and do not show any obvious abnormalities. The expression of the sponge in miR-9-Sp$^{Nestin-Cre}$ mice was confirmed by in situ hybridization, quantitative PCR, and immunodetection of GFP (*Figure 3A–E*), indicating that the expression level of the sponge transgene is sufficiently high to overcome a full repression by endogenous miR-9 activity.

Despite the fact that sponges reduce miRNA activity by sequestration, we observed that miR-9 abundance was also reduced in miR-9-Sp$^{Nestin-Cre}$ mice (*Figure 3F*), probably due to increased degradation of miR-9 in the presence of a FC target (*Ameres et al., 2010*). No compensatory increase in miR-9 precursors levels was observed (*Figure 3G*).

To analyze dendrite arborization in vivo, miR-9-Sp$^{Nestin-Cre}$ mice were crossed with Thy1$^{EGFP}$ mice (*Feng et al., 2000*). We prepared hippocampal slices from 3-month-old animals, and traced individual CA1 neurons using Neurolucida software. Pyramidal CA1 neurons from miR-9-Sp$^{Nestin-Cre}$ mice exhibited a reduction in total dendritic length and arborization complexity compared to control littermates (miR-9-Sp$^{fl-Stop}$) (*Figure 4A–C*). Similar results were obtained when neurons were visualized by Golgi staining (*Figure 4—figure supplement 1*). Importantly, this phenotype was not restricted to hippocampal neurons and it was also observed in layer V cortical neurons (*Figure 4—figure supplement 2*), suggesting that the reduction in dendritic arborization might be a generalized effect of miR-9 loss of function.

The reduction in the dendritic complexity observed in miR-9-Sp$^{Nestin-Cre}$ mice might be influenced by changes in cell-extrinsic factors of the extracellular milieu, glial cells, or decreased activity of neuronal inputs affecting dendritic differentiation (*Polleux et al., 2000*; *Horch and Katz, 2002*; *Whitford et al., 2002*; *Wong and Ghosh, 2002*; *Yamamoto et al., 2006*). To address this question, we expressed the GFP-FC-miR-9 sponge in a subset of sparse CA1 neurons by in utero electroporation. Hippocampal neural precursors of E14.5 wild-type embryos were transfected with a plasmid expressing mRFP (volume marker) together with either a GFP-FC-miR-9 expressing construct or a GFP spongeless control plasmid. Total dendritic length and dendrite complexity were analyzed 1 month after birth. In utero gene transfer of GFP-FC-miR-9-sponge plasmids resulted in changes comparable to those observed in miR-9-Sp$^{Nestin-Cre}$ mice. These results demonstrate that the dendritic changes observed with stable ablation of miR-9 are essentially cell autonomous and not caused by changes of cell extrinsic factors (*Figure 4D–F*).

Alterations in dendrite complexity occurring during development might induce structural or functional compensatory mechanisms at the synaptic level in order to preserve the functionality and computational power of the postsynaptic neuron (*Kolb et al., 1997*; *Dieni and Rees, 2003*). We first investigated dendritic spine density of CA1 hippocampal neurons in miR-9 Sp$^{Nestin-Cre}$ Thy1$^{EGFP}$ mice. No effects were observed in spine density or morphology compared to littermate controls (miR-9-Sp$^{fl-Stop}$) (*Figure 4G*). To assess the strength of synaptic transmission in pyramidal hippocampal neurons, we performed field recordings in acute hippocampal slices and analyzed the neural transmission at Schaffer collaterals-CA1 synapses. The size of the presynaptic fiber volley (FV) was compared with the

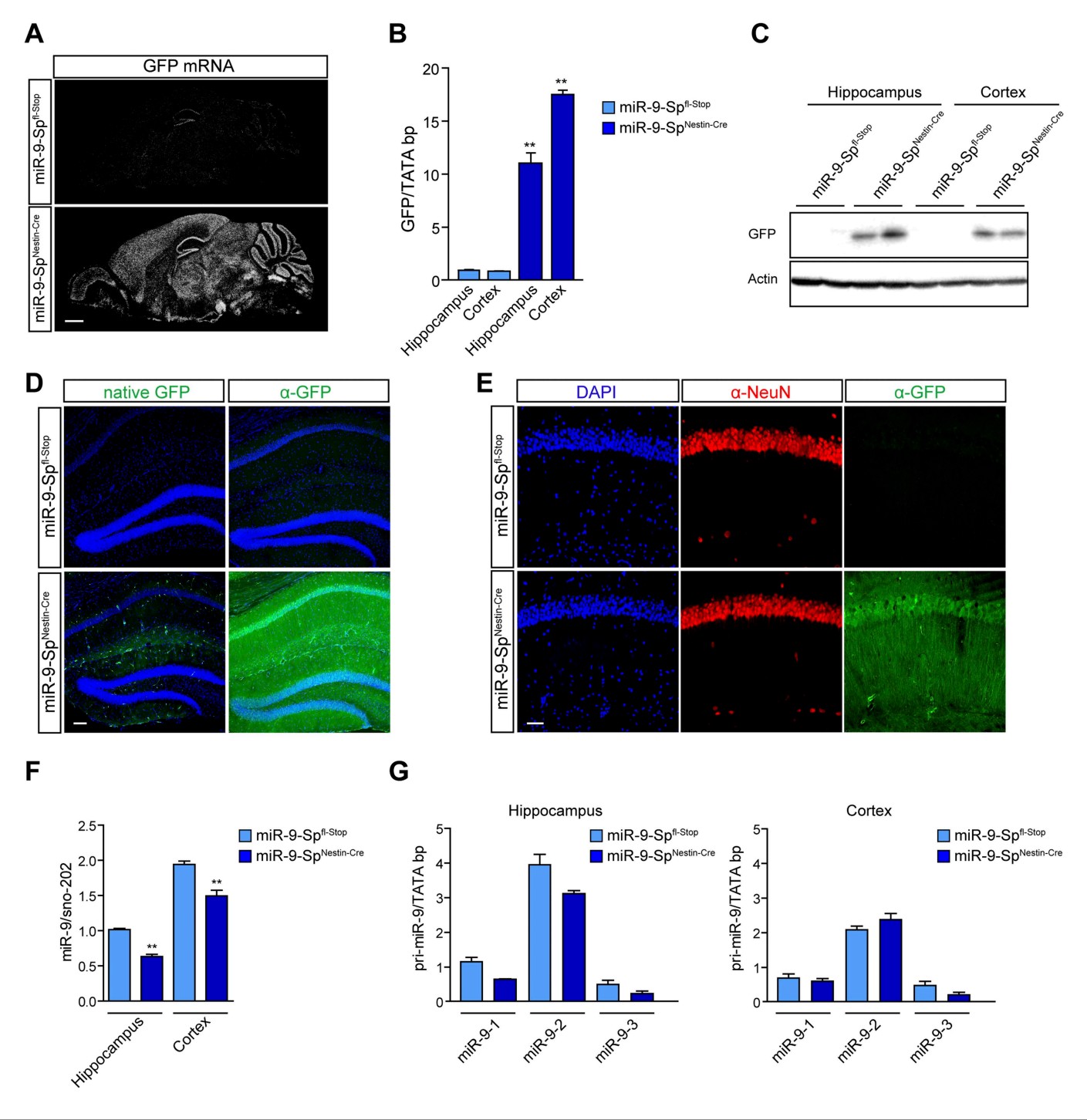

**Figure 3**. Characterization of miR-9-Sp[Nestin-Cre] mouse line. miR-9-Sp[fl-Stop] littermates were used as controls. The expression of the sponge construct was evaluated by (**A**) in situ hybridization using anti-GFP riboprobe and (**B**) qPCR. As an additional readout of the expression of the sponge, GFP protein content was determined by (**C**) Western blotting and (**D** and **E**) immunohistochemistry with anti-GFP antibodies. (**D**) Note that only a faint native GFP fluorescence is observed in miR-9-Sp[Nestin-Cre] brain sections, whereas a strong GFP signal can be detected after immunofluorescence, indicating that GFP protein levels are low but not completely down-regulated by miR-9. Scale bar: 100 µm. (**E**) Co-immunohistochemistry of GFP and NeuN, showing GFP expression in CA1 neurons. Scale bar: 50 µm. (**F**) Quantification of miR-9 levels in hippocampal and cortical samples by qPCR using Taqman probes. (**G**) Quantification of miR-9 precursors (miR-9-1, miR-9-2, and miR-9-3) in hippocampal and cortical samples by qPCR. Values represent mean + SEM, one-way ANOVA, and Bonferroni post-test, *p < 0.05, **p < 0.01, n = 4.

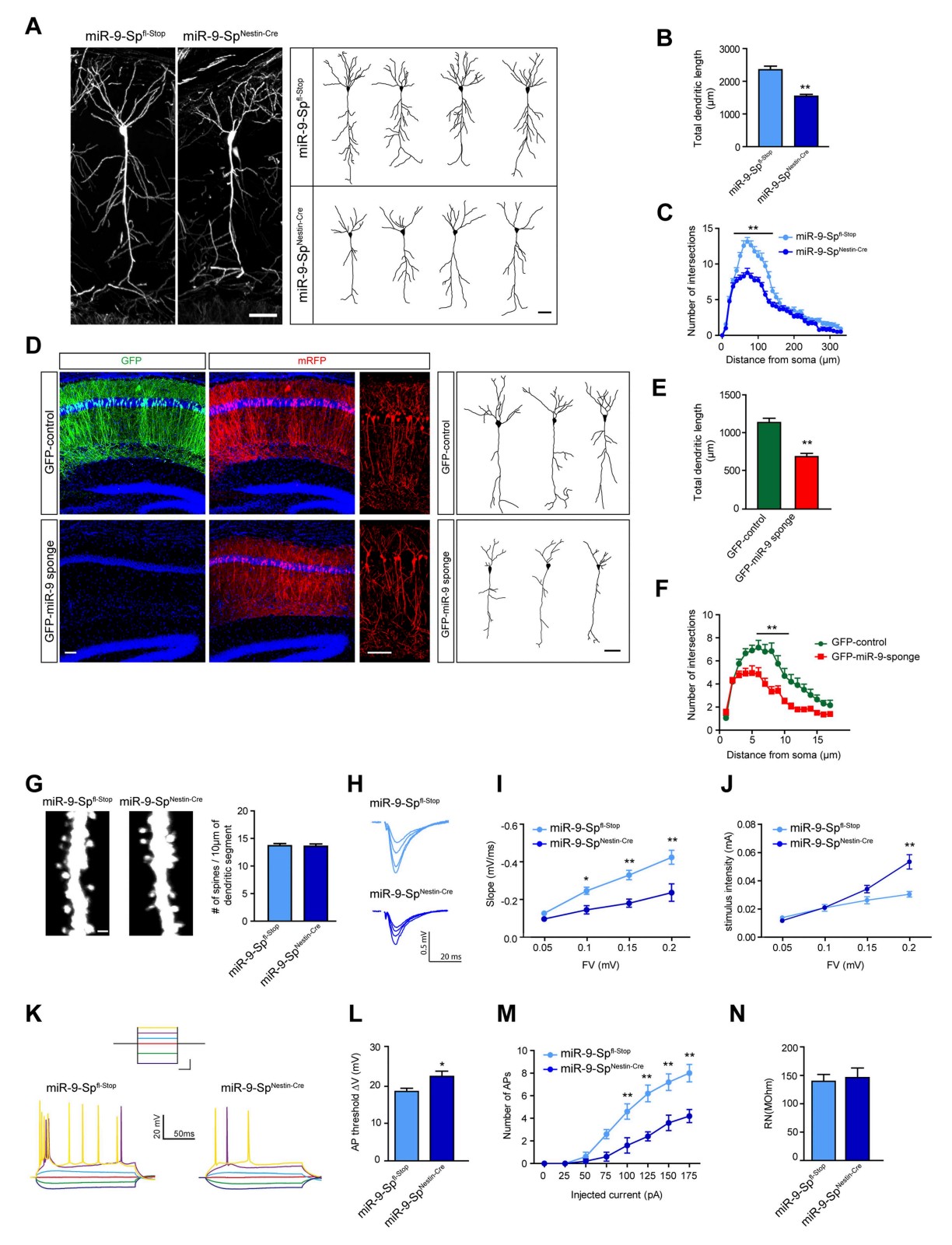

**Figure 4**. miR-9 sponge transgenic mice exhibit dendritic growth defects and impaired synaptic transmission. (**A**) Confocal stack images (left) and representative tracings (right) of CA1 neurons of Thy1$^{EGFP}$-miR-9-Sp$^{fl-Stop}$ (control) and Thy1$^{EGFP}$-miR-9-Sp$^{Nestin-Cre}$ mice. Scale bar: 50 μm. (**B**) Quantification of the total dendritic length of neuronal tracings (mean + SEM, Student's *t* test), (**C**) Sholl analysis of neuronal tracings (mean + SEM, two-way ANOVA

*Figure 4. Continued on next page*

*Figure 4. Continued*

and Bonferroni post-test). (**B** and **C**) \**p < 0.01, n ≥ 30 neurons per condition recruited from five mice per genotype. (**D**–**F**) GFP-miR-9 sponge or GFP (control) plus mRFP plasmids (for visualization) were in utero electroporated at E14.5. At P30, pyramidal hippocampal neurons in CA1 were traced with the Neurolucida software. (**D**) Images of transfected brain sections (left). Note the downregulation of GFP encoded in the miR-9 sponge plasmid. Scale bar: 100 μm. Representative tracings (right). Scale bar: 50 μm. (**E**) miR-9 sponge-transfected neurons show a reduced total dendritic length (mean + SEM, Student's *t* test) (**F**) and reduced dendritic complexity (mean + SEM, two-way ANOVA and Bonferroni post-test). (**E** and **F**) \**p < 0.01, n ≥ 18–20 neurons per condition. (**G**) Dendritic spines on secondary apical dendrites of CA1 neurons of Thy1^EGFP^-miR-9-Sp^fl-Stop^ (control) and Thy1^EGFP^-miR-9-Sp^Nestin-Cre^ mice and quantification of spine density (n.s, Student's *t* test, n ≥ 30 neurons per condition recruited from five mice per genotype). Scale bar: 1 μm. (**H**–**J**) fEPSP recordings in acute hippocampal brain slices demonstrate an impairment of synaptic transmission. (**H**) Sample traces. (**I**) Input–output curve showing reduced synaptic transmission and (**J**) reduced excitability in miR-9-Sp^Nestin-Cre^ slices (mean ± SEM, two-way ANOVA and Bonferroni post-test, \*p < 0.05, \**p < 0.01, n ≥ 9 recordings per condition recruited from 4 mice per genotype). (**K**–**N**) Current-clamp recordings. (**K**) Sample traces of current-clamp recordings. Inset shows the current step protocol, and the same colors indicate same injected currents: dark blue, −100 pA; green, −50 pA; red, 0 pA; light blue, 25 pA; magenta, 50 pA; yellow, 75 pA. Calibration (for inset): 250 ms, 50 pA. (**L**) Action potential (AP) threshold is shifted to more positive membrane potentials in CA1 neurons form miR-9-Sp^Nestin-Cre^ mice (mean + SEM, Student's *t* test) and (**M**) fewer APs were elicited with the same current injected (mean ± SEM, two-way ANOVA and Bonferroni post-test, \*p < 0.05, \**p < 0.01, n ≥ 9 recordings per condition recruited from four mice per genotype). (**N**) Input resistance was not changed (mean + SEM, Student's *t* test, n.s).

The following figure supplements are available for figure 4:

**Figure supplement 1**. miR-9 loss of function induces dendritic growth defects.

**Figure supplement 2**. Dendritic defects in cortical neurons of miR-9 sponge transgenic mice.

**Figure supplement 3**. sEPSC and mEPSC are affected by miR-9 loss of function.

slope of the EPSP in the *stratum radiatum*. We found that synaptic transmission was significantly reduced in miR-9-Sp^Nestin-Cre^ mice compared to miR-9-Sp^fl-Stop^ littermate controls (**Figure 4H,I**). Interestingly, during the course of our experiments, we noted that higher stimulus intensities were required in miR-9-Sp^Nestin-Cre^ mice to elicit the same FV amplitude (**Figure 4J**). This prompted us to directly test the excitability of CA1 pyramidal cells. In current-clamp recordings we observed a shift of the action potential (AP) threshold to more positive membrane potentials in miR-9-Sp^Nestin-Cre^ neurons (**Figure 4K,L**). In turn, the same current injected elicited fewer APs (**Figure 4M**). The input resistance was not changed (**Figure 4N**). If pyramidal neurons are less excitable and therefore generate fewer APs, we should observe a reduced frequency of spontaneous EPSCs (sEPSC). Indeed, the mean sEPSC frequency was largely reduced in CA1 neurons of sponge-expressing animals (**Figure 4—figure supplement 3A,C**). The reduced sEPSC amplitude (**Figure 4—figure supplement 3B**) could be caused by the reduced number of AP driven events; however, to test whether the quantal size is reduced in miR-9-Sp^Nestin-Cre^ mice, we recorded miniature EPSCs (mEPSC). Here, mEPSC amplitude and frequency were significantly reduced (**Figure 4—figure supplement 3D–F**). The reduced frequency reflects the reduced number of synapses, while the reduced mEPSC amplitude indicates a change in quantal size. To exclude presynaptic effects, we tested paired-pulse facilitation, a sensitive measure of changes in the probability of transmitter release, which was unchanged (**Figure 4—figure supplement 3G**). Taken together, these results suggest that the impaired synaptic transmission is caused by a reduction in the absolute number of synapses per neuron (with preserved synapse density), resulting from the shortened dendrites in miR-9-Sp^Nestin-Cre^ mice. Additionally, inhibiting miR-9 affects neuronal excitability by changing the AP threshold.

## Dendritic growth defects after miR-9 loss are mediated by REST

The cell-intrinsic nature of the effects of the GFP-FC-miR-9 sponge indicates that miR-9 regulates neuronal targets primarily involved in the process of dendritogenesis. To gain further insight into the mechanisms by which miR-9 affects dendritic growth, we used quantitative PCR to determine the expression levels of several predicted miR-9 neuronal targets (**Supplementary file 1**). To avoid contaminations by other cell types, we analyzed DIV3 primary hippocampal neurons derived from miR-9-Sp^Nestin-Cre^ and littermate control embryos.

We found that 13 out of 26 miR-9 predicted targets were upregulated in miR-9-Sp^Nestin-Cre^ neurons (**Figure 5A**). Interestingly, one of the upregulated mRNAs encodes the global transcriptional repressor

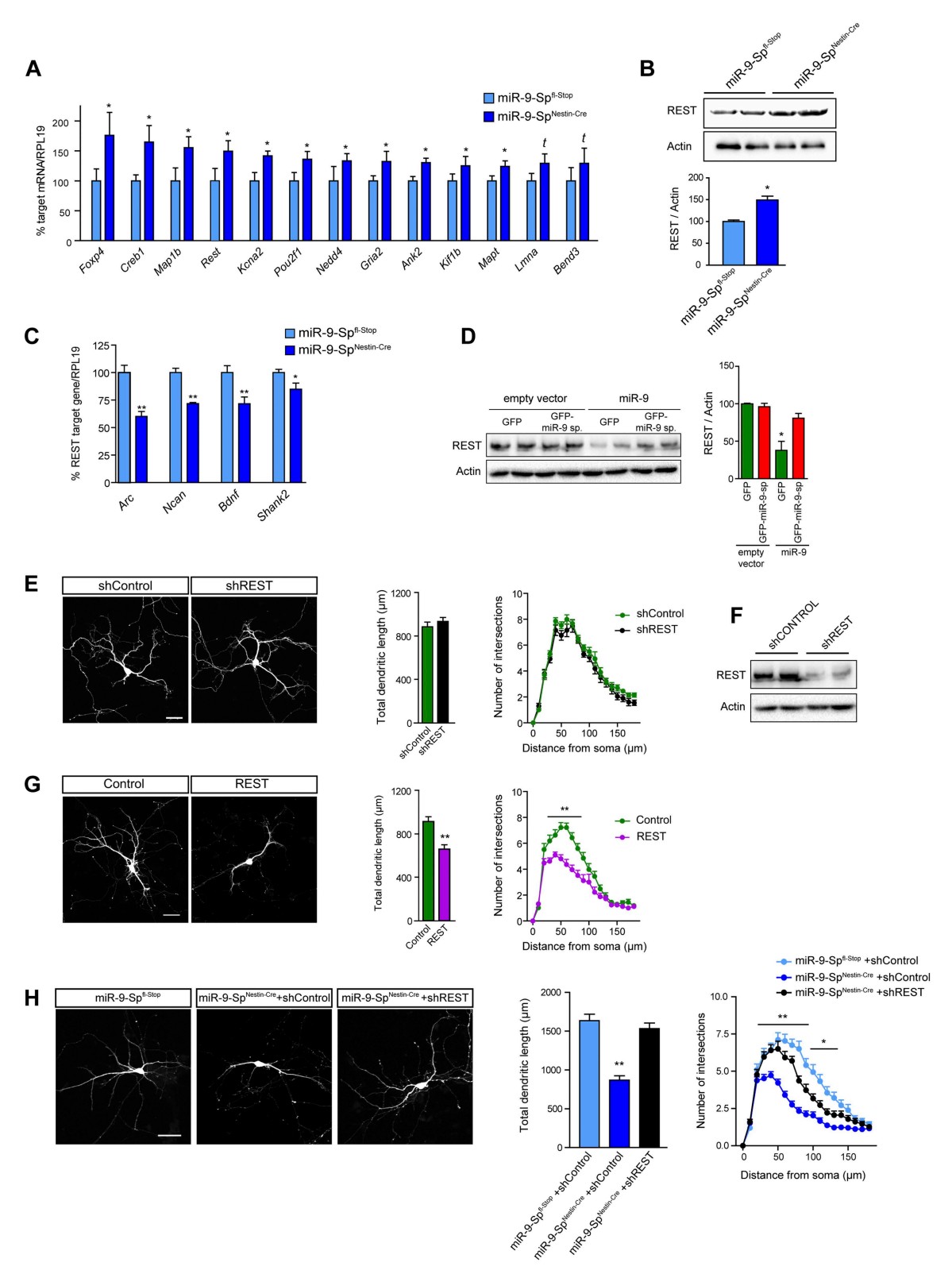

**Figure 5**. Dendritic growth defects after miR-9 loss are mediated by REST. (**A**) Quantitative PCR analysis of predicted miR-9 target mRNAs in hippocampal primary neurons (DIV3) prepared from miR-9-Sp$^{Nestin-Cre}$ and miR-9 Sp$^{fl-Stop}$ (control) E17.5 embryos (mean + SEM, paired Student's *t* test, n = 4, *p<0.05; *Supplementary file 1*). (**B**) Representative immunoblot and quantification of the normalized OD (mean + SEM, Student's *t* test, n = 3, *p < 0.05).
*Figure 5. Continued on next page*

*Figure 5. Continued*

(**C**) Quantitative PCR analysis of the expression of REST targets in the same neurons as in (**A**) (mean + SEM, paired Student's *t* test, n = 4, *p < 0.05; **p < 0.01). (**D**) Representative immunoblot (left) and quantification (right) showing that FC-miR-9 sponges reversed miR-9-induced silencing of endogenous REST in transfected HEK-293 cells. Note that REST levels are not upregulated in the condition 'empty vector + GFP-miR-9 sponge', because miR-9 is not expressed in HEK-293 cells (mean + SEM, one-way ANOVA, and Bonferroni post-test, *p < 0.05, n = 3.). (**E**) REST knock-down has no effect on dendritic length and complexity. Primary hippocampal neurons were transfected at DIV4 with shRNAs against REST (shREST) or shControl. (**F**) Efficacy of shREST vectors. Representative immunoblot showing the downregulation of endogenous REST induced by shRNAs against REST in transfected HEK cells. (**G**) REST overexpression reduces dendritic length and complexity in primary neurons. Primary hippocampal neurons were transfected at DIV4 with REST or empty vector (control). (**H**) Primary hippocampal neurons were transfected via ex vivo electroporation. (**E**, **G**, **H**) Scale bar 50 µm. RFP was cotransfected for visualization. Values represent mean + SEM, Student's *t* test (dendritic length) or two-way ANOVA and Bonferroni post-test (Sholl analysis), *p < 0.05, **p < 0.01. (**E** and **G**) n ≥ 20 neurons per condition. (**H**) n ≥ 30 neurons per condition.
The following figure supplement is available for figure 5:

**Figure supplement 1**. Increased levels of Foxp4, Creb, or Map1b have no impact on dendritic length and complexity.

element 1 (RE-1) silencing transcription factor (REST/NRSF). We speculated that REST is an interesting candidate, which could explain the effects triggered by GFP-FC-miR-9 sponge considering that REST (i) is actively repressing a large array of neuron-specific genes (***Ballas and Mandel, 2005***), (ii) is strongly downregulated during neuronal development (***Schoenherr and Anderson, 1995***; ***Ballas et al., 2005***), and (iii) has been experimentally validated as a bona fide target of miR-9 (***Packer et al., 2008***). Importantly, we also found elevated REST protein levels in sponge expressing neurons (***Figure 5B***). As expected, increased REST activity resulted in the downregulation of several REST-target genes (***Figure 5C***). Moreover, miR-9 sponges reversed the miR-9-induced downregulation of REST in HEK-293 cells (***Figure 5D***) further demonstrating the regulatory effect of miR-9 on REST levels.

To fulfill the criteria as a potential mediator of miR-9-dependent effects on dendritic arborization in miR-9-Sp^Nestin-Cre mice, REST should negatively regulate dendritic growth. However, this is not evident from REST loss-of-function models. Conditional deletion of *Rest* in early neural progenitors in *Rest*^Sox1–Cre mice, did not affect brain histology and had no impact on the expression of REST-target genes (***Aoki et al., 2012***). In agreement, we found no difference in dendritic development between shREST- and shControl-transfected neurons (***Figure 5E***). The efficiency of shREST constructs to downregulate REST was assessed in HEK-293 cells (***Figure 5F***). Together with the lack of phenotype observed in *Rest*^Sox1–Cre mice, these results suggest that the transcriptional repressive activity of endogenous REST is minimal in neurons. Since REST is strongly downregulated during neuronal development, we next hypothesized that maintenance of high levels of this transcriptional repressor might be detrimental for neuronal development and that, consequently, a key function of miR-9 would be to keep low REST levels across neuronal maturation. In agreement with this hypothesis, we found that overexpression of REST in primary hippocampal neurons significantly reduces dendritic length and arborization (***Figure 5G***). In contrast, overexpression of other miR-9 targets that were found to be upregulated in sponge expressing neurons, such as Foxp4, Creb1, and Map1b, had no effect on dendritic length and complexity (***Figure 5—figure supplement 1***), indicating that they are not contributing to the dendritic phenotype of miR-9-Sp^Nestin-Cre neurons.

To test whether REST upregulation mediates the impaired dendritic growth in sponge mice, we transfected neural precursors with shRNAs against REST by ex vivo electroporation. In this approach, miR-9-Sp^Nestin-Cre and littermate control embryos were decapitated at E17.5, DNA was injected into lateral ventricles followed by electroporation targeting hippocampal neural precursors directly before dissociated primary hippocampal neurons were prepared. We observed that shREST transfection strongly rescued the dendritic growth defects in miR-9-Sp^Nestin-Cre neurons (***Figure 5H***), indicating that increased REST level is indeed the major mediator of miR-9 effects on dendritic growth.

## Discussion

The results of this study define a key role for miR-9 during postmitotic differentiation of neurons.

We present a new double reporter mouse line that helped to characterize the activity pattern of miR-9 in different cell types in the brain. The analysis of the miR-9-Sp^Nestin-Cre mouse line showed that inhibition of miR-9 results in aberrant dendritic growth in vivo, accompanied by a concomitant impairment in synaptic transmission. Experiments using primary neurons and in utero gene transfer approaches

indicate that the observed dendritic changes are cell-autonomous. Importantly, we described an essential function of miR-9 in dendritic growth as a regulator of the expression of the transcriptional repressor REST.

The widespread involvement of miRNAs in both the regulation of developmental and physiological processes and disease is well established (*Bushati and Cohen, 2007*; *Bartel, 2009*; *Han et al., 2013*). However, uncovering the function of specific microRNA families has been limited by the lack of technologies for in vivo analysis. Much of our current knowledge regarding the role of microRNAs in neuronal biology stems from overexpression-based studies. However, overexpression approaches may lead to silencing of genes that normally escape regulation resulting in artificial, neomorphic phenotypes. Therefore, loss-of-function strategies are required to validate physiological miRNA functions. The generation of KO mice has several limitations in this context: (i) requires the ablation of the multiple gene copies of miRNAs belonging to the same family, often placed at distant genomic loci and (ii) miRNAs genes are frequently encoded in clusters together with other miRNAs from different families or directly overlapping other protein-coding or lncRNA genes. Therefore, the ablation of the miRNA sequence by gene targeting KO strategies also affects the expression of the 'host' genes coding for proteins or lncRNAs, or miRNA genes from other families encoded in tandem; and (iii) the complementary miRNA* strand is entirely removed by the gene deletion.

In the case of miR-9, recently generated *miR-9-2*/*miR-9-3* double KO mice showed profound phenotypic brain defects, growth retardation, and reduced life-span (*Shibata et al., 2011*). In this model, both miR-9 and miR-9* levels are reduced. Unlike other mRNA* species, miR-9* accumulates to substantial levels in vivo (*Bak et al., 2008*; *Yao et al., 2012*), and since miR-9* has a different seed sequence compared to miR-9, it is predicted to regulate a different set of targets. Importantly, miR-9* regulation of BAF53a has been shown to contribute to activity-dependent dendritic growth (*Yoo et al., 2009*). In addition, two transcripts (C130071C03Rik-001 and C130071C03Rik-201) that overlap with the murine *miR-9-2* gene and encode for lncRNAs, are deleted in this mouse model. Although the function of these transcripts is still unknown, recent evidence demonstrates that ablation of lncRNA genes can produce severe phenotypes at brain level (*Sauvageau et al., 2013*), indicating that they would play more critical roles in vivo than previously considered. In view of these facts, other functional products in addition to miR-9 might contribute to the phenotype of *miR-9-2*/*miR-9-3* KO mice and further investigations will be required to clarify this issue. In contrast, the use of miR-9 sponges as dominant negative genetic tools, allowed us to dissect the specific role of miR-9 on dendritic and spine development. Previously, miRNA sponges have been proven as useful genetic loss-of-function tools in *Drosophila* (*Loya et al., 2009*), and non-conditional expression of sponges has been reported in mice (*Ma et al., 2011*). Here, we provide a new layer of complexity to the existing battery of in vivo tools for genetic manipulation of microRNAs by describing the first conditional transgenic miRNA sponge mouse line.

We initially tested the efficiency of FC vs bulged miR-9 sponges and found that both designs were equally effective to inhibit miR-9 activity. Although several studies describe efficient inhibitory activity of FC sponges (*Care et al., 2007*; *Ebert et al., 2007*; *Gentner et al., 2009*), a recent study by Kluiver et al. (*Kluiver et al., 2012*), reported that bulged sponges are more effective. The discrepancy with our results could arise from differences in promoter strengths. We employed a CAG promoter for the sponge constructs instead of the PGK promoter used in the referred study. The CAG promoter is stronger than the PGK in most mammalian cell systems (*Qin et al., 2010*; *Chen et al., 2011*) and in the Neuro-2a cells used in this study (data not shown). Thus, high levels of expression of the sponge achieved in our experiments could compensate the degradation of FC sponge transcript due to endonucleolytic cleavage activity of AGO2, resulting in high steady-state levels of the decoy target.

Histological approaches to study miRNA expression have often relied on in situ hybridization (ISH) techniques (*Akerblom et al., 2012*). However, with these techniques, it is not possible to discriminate between precursor and mature miRNAs. Moreover, the design of the probes is very limited due to the very short sequence of miRNAs, preventing the optimization steps usually necessary to obtain adequately specific and sensitive riboprobes for ISH (*Refojo et al., 2011*). In this study, we used the reporter function of miRNA sponges that have the advantage of reporting miRNA activity instead of expression. The miR-9-Sp/tdTomato mouse line was used to analyze the activity pattern of miR-9 in the adult mouse brain. We observed high miR-9 activity in neurons and astrocytes and a lower activity in microglial cells. While miR-9 has been detected in microglia (*Liu et al., 2013a*) and other cells of myeloid origin (*Senyuk et al., 2013*), findings obtained with a recently developed miR-9 reporter

mouse line, miR-9.T, suggest that miR-9 activity is absent in microglia (*Akerblom et al., 2013*). Lower levels of the PGK-driven GFP-miR-9 sensor transgene may account for the lower sensitivity shown by that mouse model. Although the expression of the sensor is not shown outside the brain, an indication of the low expression of the sensor of miR-9.T mice is that GFP expression could only be analyzed after the enhancement of the signal via immunofluorescence (*Akerblom et al., 2013*), in contrast to our characterization that fully relies on the detection of the native fluorescence of GFP.

The function of miR-9 has been mainly studied in the context of neurogenesis, where it plays a role as a regulator of the balance between progenitor proliferation and neuronal differentiation (*Leucht et al., 2008*; *Zhao et al., 2009*; *Delaloy et al., 2010*; *Bonev et al., 2011*; *Coolen et al., 2012*). These studies showed that miR-9 does not act as a differentiation switch, but rather facilitates the transition from the proliferative to the neurogenic mode (*Coolen et al., 2012, 2013*). In contrast, only few studies have unraveled additional functions of miR-9 at later steps of neural development. A role of miR-9 on the specification of spinal cord motoneurons has been described (*Otaegi et al., 2011*; *Luxenhofer et al., 2014*). MiR-9 dysregulation has also been linked to spinal motor neuron disease (*Haramati et al., 2010*). Recently, *Dajas-Bailador et al. (2012)* showed that inhibition of miR-9 in mouse cultured cortical neurons increase axonal length and reduce axonal branching. In this study, we focused in the postmitotic stages of neuronal differentiation and demonstrated a critical role for miR-9 in the control of dendritic development in vivo. Other miRNAs have also been shown to influence dendritic growth (*Wayman et al., 2008*; *Fiore et al., 2009*; *Parrish et al., 2009*; *Magill et al., 2010*; *Pathania et al., 2012*; *Liu et al., 2013b*) suggesting that the expression of genes orchestrating this process is under a complex post-transcriptional regulation.

Our studies in cultured primary neurons associated to the morphological analysis performed in miR-9-Sp[Nestin-Cre] mice suggests that neuronal miR-9 mainly acts in a cell-intrinsic manner controlling dendritic differentiation genetic programs. However, dendritic growth and branching in vivo are influenced by synaptic activity (*Rajan and Cline, 1998*; *Wong and Ghosh, 2002*) or by a multiplicity of extrinsic factors either present in or secreted by neighbor neurons or glial cells, such as neurotrophins, semaphorins, or Slits (*Polleux et al., 2000*; *Horch and Katz, 2002*; *Whitford et al., 2002*; *Yamamoto et al., 2006*).

The results obtained in our in utero gene transfer studies targeting CA1 hippocampal neurons, which develop later among a vast majority of non-transfected wild-type neurons and glial cells, are comparable to the dendritic phenotype observed in miR-9-Sp[Nestin-Cre] mice. This indicates that effects found in miR-9-Sp[Nestin-Cre] mice are essentially cell-intrinsic and that miR-9 exerts its proactive function in modeling dendritic arbors as a key player of neuronal genetic programs controlling dendrite development. Furthermore, we found that basal synaptic transmission was impaired in miR-9 sponge expressing mice. Since the morphology and density of dendritic spines were not affected, the reduced postsynaptic responses are most likely due to a reduction in the total number of synapses as expected from less-arborized dendrites (*Kawabe et al., 2010*), as those found in miR-9-Sp[Nestin-Cre] mice. Additionally, we showed that inhibiting miR-9 affects neuronal excitability by shifting the AP threshold to more positive potentials. Since Map1b has already been described to be a miR-9 target critical for terminal axonal arborization (*Dajas-Bailador et al., 2012*), a potential contribution of the tandem miR-9/Map1b to the electrophysiological phenotype cannot be discarded.

Intrinsic proteins controlling the establishment of neuronal dendritic fields include several transcription factors, Rho and Ras family GTPases, kinases, and adaptors involved in $Ca^{2+}$-mediated signal transduction cascades (*Hand et al., 2005*; *Takemoto-Kimura et al., 2007*; *Simo and Cooper, 2012*). MiR-9 downregulation would lead to the increase in transcriptional rates of direct miR-9 targets. This suggests that the factor/s mediating the effects of miR-9 inhibition in miR-9-Sp[Nestin-Cre] mice would fit best with targets negatively influencing cellular processes potentially involved in dendritic maturation. We found that the zinc-finger transcriptional repressor REST (also known as NRSF), an experimentally validated target of miR-9 (*Packer et al., 2008*), is upregulated in miR-9 sponge expressing neurons. Along this line, the fact that REST is a transcriptional repressor made this factor a specially attractive candidate to explain the effects observed in miR-9-Sp[Nestin-Cre] mice.

REST is essential for repressing neuronal genes in neural progenitors (*Ooi and Wood, 2007*). A prevailing view is that downregulation of REST during neuronal differentiation favors the acquisition and maintenance of the neuronal phenotype (*Ballas et al., 2005*). REST binds the RE1 consensus element of target genes and recruits corepressors including mSin3, CoREST, and methyl CpG binding protein 2 (MeCP2), inducing silencing by epigenetic remodeling (*Andres et al., 1999*).

The influence of REST on the dendritic development of pyramidal neurons remains unexplored. Although a mild effect on neurite outgrowth was reported in cultured retinal ganglion cells upon titration of REST with sequestering plasmids (*Koch et al., 2011*), we were unable to observe any effect of REST downregulation on the dendritic length or complexity of hippocampal pyramidal neurons during development. In agreement with our results, the conditional deletion of *Rest* in early neural progenitors had no impact on the expression of REST-target genes in postnatal day 0 brains, when dendrite differentiation is taking place (*Aoki et al., 2012*). Together, these results suggest that the repressive activity of endogenous REST is minimal during postmitotic differentiation of neurons.

Considering that REST is strongly downregulated during neuronal development, we hypothesized that maintenance of high levels of this transcriptional repressor might be detrimental for neuronal development. Therefore, a key function of miR-9 would be to keep low REST levels across neuronal maturation. Consistent with this hypothesis, we show that REST overexpression in primary neurons has a negative impact on dendritic length and complexity. Importantly, by performing ex vivo gene transfer of REST shRNAs into neural precursors, we determined that the increased REST activity in miR-9 sponge neurons is underlying the impairment in dendritic growth. However, the fact that the rescue of the dendritic defects was extensive but not complete, suggests that other miR-9-regulated proteins may also contribute to the dendritic phenotype of miR-9 sponge expressing neurons. Together, these experiments delineate a central mechanism in which miR-9 restricts REST expression and thereby promotes neuronal postmitotic differentiation. Interestingly, it has been observed that REST levels are restored in some neuronal populations following a biphasic expression (*Gao et al., 2011*). Consequently, REST expression have been shown to be increased in some adult hippocampal and cortical neurons (*Kuwabara et al., 2004*), suggesting that miR-9/REST regulation might play other non-developmental roles in some mature neuronal populations.

In summary, we describe the first transgenic sponge mouse line that allows for conditional inactivation of an miRNA family in a spatio-temporal-controlled manner. Analysis of this miR-9-Sp[Nestin-Cre] mouse line reveals a prominent role for miR-9 on postmitotic neuronal differentiation as a regulator of dendritic development through repression of REST expression.

## Materials and methods

### Animals

In all experiments, mice were housed under standard laboratory conditions (22 ± 1°C, 55 ± 5% humidity) with food and water ad libitum. For staging of embryos, noon on the day of the appearance of a vaginal plug was treated as embryonic day 0.5 (E0.5), and the day of birth was considered postnatal day 0 (P0). Animals were handled according to the Guide for the Care and Use of Laboratory Animals of Government of Bavaria, Germany.

### DNA constructs

The CAG-GFP-sponge-bGH-polyA plasmids were self-constructed using a modified pCRII vector as backbone (Life Technologies, Darmstadt, Germany). Sponges consisting of 18 repetitive sequences separated by a variable four-nucleotide linker were chemically synthesized (GenScript Corp., Piscataway, NJ, USA) and introduced in the 3'UTR of GFP cDNA. Individual sequences for each sponge are as follows: FC miR-9 sponge 5'-TCAT-ACAGCTAGATAACCAAAGA-3'; Bg miR-9 sponge 5'-TCATACAGCTATATACCAAAGA-3'; scrambled miR-9 sponge 5'-ATG-ATAACAACGAACGATTACAC-3'; FC miR-9* sponge 5'-ACTTTCGGTTATCTAGCTTTAT-3'; Bg miR-124 sponge 5'-GGCATTCACAAGTGC CTTA-3'. The control spongeless plasmid pCAG-GFP was subsequently generated by removing the sponge sequence and religated after adding a short linker. Overexpression of miR-9 was performed with the vector pFhSynW-mCherry-pri-miR-9 generously provided by D Edbauer (*Edbauer et al., 2010*). Based on that construct, we generated an overexpression construct for mutated miR-9 (5'-U<u>AACGACG</u>UAUCUAGCUGUAUGA-3') using the QuikChange Site-Directed Mutagenesis Kit (Stratagene, La Jolla, CA). To generate the luciferase miR-9 sensor, the firefly luciferase cDNA was cloned into the pCRII vector containing the CAG promoter described before. A short sequence containing a single fully complementary binding site to miR-9 was added as 3'UTR. In luciferase assays, a CMV-lacZ plasmid was cotransfected to normalize transfection efficiency. The plasmid pHR'-NRSF-CITE-GFP (Addgene plasmid 21310) (*Nadeau and Lester, 2002*) was used to overexpress REST (insensitive to miR-9 regulation since lacks 3'UTR). To downregulate REST, we used a mixture

of two shRNA constructs generously shared by A Fisher (*Jorgensen et al., 2009*) who also provided the shControl construct. A strong downregulation of REST levels was observed following transfection of shREST constructs in HEK cells (*Figure 5F*). The plasmid CAG-Map1b, used to overexpress Map1b, was generously provided by T Kawauchi (*Kawauchi et al., 2005*). To overexpress Foxp4, Foxp4 mouse cDNA was cloned into a pcDNA3 expression vector. The targeting vector to generate miR-9 sponge transgenic mice was based on pROSA26-1 bearing 5.5-kb homology to the murine *ROSA26* locus (*R26*) (*Soriano, 1999*). It was cloned by introducing the following components: *frt* site, CAG prmoter, *frt* site, *loxP* site, STOP cassette, *loxP* site, EGFP, FC miR-9 sponge, bGH polyadenylation sequence (pA) (from 5′ to 3′) (*Figure 2—figure supplement 1*).

## Generation of conditional GFP-FC-miR-9 sponge mice and derived mouse lines

The linearized targeting vector was electroporated into mouse F1 embryonic stem (ES) cells (IDG 3.2) (*Hitz et al., 2007*). Mutant ES cell clones were identified by Southern blot analysis of genomic ES cell DNA. Mutant ES cells were used to generate chimeric mice by blastocyst injection. Germ-line transmission of the modified *R26* (miR-9-Sp[fl-Stop]) allele was confirmed in offsprings from male chimeras bred to wild-type C57BL/6J mice. Conditional, CNS-restricted expression of the GFP-FC-miR-9 sponge transcript was achieved by breeding miR-9-Sp[fl-Stop] mice with transgenic Nestin-Cre mice (*Tronche et al., 1999*).

To generate the miR-9 reporter mouse line, miR-9-Sp[fl-Stop] and tdTomato[fl–Stop] (Ai9 line *Madisen et al., 2010*) homozygous mice were bred with a Deleter-Cre mouse line (TaconicArtemis GmbH, Köln, Germany) carrying an ubiquitously expressed Cre transgene in the *R26* locus. This leads to a removal of the 'Stop' cassette from the miR-9-Sp[fl-Stop] and the tdTomato[fl–Stop] alleles (both also in the *R26* locus) in all tissues of F1 mice, including the germ line. Resulting heterozygous miR-9-Sp/Deleter-Cre and tdTomato/Deleter-Cre F1 animals were intercrossed to obtain in the F2 generation miR-9-Sp/tdTomato reporter mice (*Figure 2—figure supplement 2*).

## Culture of cell lines and transient transfections

Neuro-2a and HEK293 cells were grown in Dulbecco's modified Eagle's medium (DMEM) supplemented with 10% heat-inactivated FCS, Penicillin (100 units/ml), Streptomycin (100 μg/ml), and 2 mM L-Glutamine (Gibco, Kalsruhe, Germany) at 37°C and 5% $CO_2$. For transient transfection experiments, cells were seeded at a desired density on poly-D-Lysine (0.05 mg/ml, 30.000–70.000 MW, Sigma Aldrich)-coated cell culture plates. 24 hr later cells were transfected with the respective expression vectors using Lipofectamine 2000 (Invitrogen, Kalsruhe, Germany).

## Primary cell cultures and transfection

Primary hippocampal neurons were prepared from CD1 or miR-9-Sp[fl-Stop]/miR-9-Sp[Nestin] mouse embryos (E17.5–18.5) and maintained in Neurobasal-A medium with 2% B27 and 0.5 mM GlutaMAX-I (Gibco) at 37°C and 5% $CO_2$ (*Dotti et al., 1988*; *Goslin et al., 1998*). Neurons were transfected via a calcium phosphate protocol (*Jiang and Chen, 2006*) unless otherwise stated. To analyze dendrite length and arborization, neurons were transfected at the DIV 4 and fixed at DIV 10. To study spine density and morphology, neurons were transfected at DIV12-14 and fixed at DIV 21.

## Luciferase and fluorometry assays

For luciferase assays, Neuro-2a cells and primary neurons were transfected as described above, and luciferase expression was analyzed in cell lysates 24 hr after transfection (Promega, Mannheim, Germany). Fluorometric quantitation of GFP was performed on cell lysates using a microplate fluorometer (Tecan Genios Pro, MTX Lab Systems Inc, Crailsheim, Germany.) with excitation and emission wavelengths of 485 nm and 535 nm respectively. Firefly luciferase activity and GFP fluorescence were normalized to β-galactosidase activity and expressed as a percentage of the control.

## Analysis of neuronal morphology

For the analysis of dendrites in culture, primary hippocampal neurons were fixed at DIV 10 and images taken from at least four independent preparations. Dendrites were traced with open access NeuronJ (ImageJ, http://rsbweb.nih.gov/ij/) and further evaluated with the same software to calculate total dendritic length and perform Sholl analysis.

For the in vivo analysis, neurons were visualized by breeding sponge mice with Thy1-EGFP mice (M) (*Feng et al., 2000*) or by performing Golgi staining with the FD Rapid GolgiStain Kit

(FD Neurotechnologies, Columbia, MD). Dendrites form CA1 pyramidal neurons or layer V neurons in the primary somatosensory cortex were traced under a 40× objective lens using the Olympus BX51 microscope installed with Neurolucida 6 (MBF Bioscience, Williston, VT). Analyses of dendritic length and Sholl analysis were evaluated with Neurolucida Explorer (MBF Bioscience). n ≥ 30 neurons per condition were analyzed, recruited from five mice per genotype.

## Electrophysiology

Acute brain slices for electrophysiological recordings were obtained from P14-P19 miR-9-Spfl-Stop and miR-9-SpNestin littermates which were anesthetized with isoflurane and decapitated. The brain was removed and chilled in ice cold carbonated artificial cerebrospinal fluid (ACSF) containing the following (in mM): 130 NaCl, 2.75 KCl, 1.43 $MgSO_4$, 1.1 $NaH_2PO_4$, 28.01 $NaHCO_3$, 2.5 $CaCl_2$, 11 glucose.

All electrophysiological recordings were performed at room temperature. Data were acquired using a Multiclamp 700B amplifier (Axon Instruments, Foster City, CA) and digitized with a Digidata 1440A (Axon Instruments).

For field recordings, Schaffer collaterals were stimulated and postsynaptic potentials recorded in the stratum radiatum of CA1. Recording and stimulation electrodes were filled with ASCF.

Whole cell recordings were obtained in ACSF supplemented with picrotoxin (100 µM) and an internal solution was used containing the following (in mM): 150 Cs-gluconate, 8 NaCl, 2 MgATP, 10 HEPES, 0.2 EGTA, and 0.5 QX-314. Miniature EPSCs (mEPSCs) were recorded at −70 mV in ACSF supplemented with TTX (0.2 µM), picrotoxin (100 µM), and trichlormethiazide (250 µM) to increase mEPSC frequency. Spontaneous EPSCs (sEPSCs) were recorded at −70 mV in ACSF supplemented with picrotoxin (100 µM) and trichlormethiazide (250 µM). mEPSCs and sEPSCs were detected offline and analyzed with a custom written MATLAB routine. Paired-pulse facilitation (PPF) was determined with a stimulus interval of 40 ms.

Current-clamp recordings were performed with a pipette solution containing the following (in mM): 150 K-methylsulfhonate, 4 KCl, 4 NaCl, 4 MgATP, 0.4 MgGTP, and 10 HEPES. Action potential thresholds (AP) were determined from the first derivative of single APs.

## In silico analysis

The binding affinity of miR-9 to the FC or Bg miR-9 sponges was predicted with RNAhybrid and miRanda software. The RNAhybrid program, Version 2.1, is available online at http://bibiserv.techfak. uni-bielefeld.de/rnahybrid/ (*Kruger and Rehmsmeier, 2006*) whereas miRanda, September 2008 release, can be accessed at http://www.microrna.org/microrna/home.do (*Witkos et al., 2011*). Values obtained with RNAhybrid are provided in the main text. Similar values were obtained with miRanda (FC sponge: 39.61 vs Bg sponge: 28.31 kcal/mol).

## In utero and ex vivo electroporation

In utero electroporation of CD1 mouse embryos was performed at E14.4 as previously described (*Saito, 2006*) with small modifications. In other experiments, hippocampal progenitors from miR-9-Sp[fl-Stop] and miR-9-Sp[Nestin] littermates were electroporated ex vivo at E17.5 as described previously (*Hand et al., 2005*). Briefly, E17.5 embryos were decapitated and heads were kept in complete HBSS buffer before use. Plasmid DNA was injected into lateral ventricles followed by electroporation with an Electro Square Porator ECM830 and tweezertrodes (BTX Genetronics, San Diego, CA). Five pulses with 40V, 50-ms duration and with 950-ms intervals, were delivered to each embryo. Directly after electroporation, primary hippocampal neuronal culture was prepared.

## In situ hybridization (*IS*H)

In situ hybridization (ISH) was performed as previously described (*Refojo et al., 2011*). Brains were carefully removed and immediately shock-frozen on dry ice. Frozen brains were cut on a cryostat in 20-µm thick sections and mounted on SuperFrost Plus slides. Specific riboprobes for GFP were generated by PCR applying T7 and T3 or SP6 primers using plasmids containing the above-mentioned cDNA as template. Radiolabeled sense and antisense cRNA probes were generated from the respective PCR products by in vitro transcription with [35]S-UTP using T7 and T3 or SP6 RNA polymerase.

Hybridization was performed overnight with a probe concentration of $7 \times 10^6$ CPM/ml at 57°C and slides were washed at 64°C in 0.1× saline sodium citrate (SSC) and 0.1 M dithiothreitol. Hybridized slides were dipped in autoradiographic emulsion (type NTB2), developed after 2 weeks and counterstained with cresyl violet.

Dark-field photomicrographs were captured with digital cameras adapted to an imaging microscope and a stereomicroscope. Images were digitalized using Axio Vision 4.5, and afterwards photomicrographs were integrated into plates using image-editing software. Only sharpness, brightness, and contrast were adjusted. For an adequate comparative analysis in corresponding miR-9-Sp$^{fl-Stop}$ (control) and miR-9-Sp$^{Nestin-Cre}$ sections the same adjustments were undertaken. Brain slices were digitally cut out and set onto an artificial black background.

## Immunoblotting experiments

Cells and tissue were lysed in RIPA buffer containing protease inhibitors (Roche, Mannheim, Germany). Protein samples were separated by 8–10% SDS-PAGE (*Laemmli, 1970*) and transferred to 0.45-µm PVDF membranes (Millipore, Darmstadt, Germany). Chemiluminescence signal was acquired in a ChemiDoc station (BioRad, Munich, Germany) and analyzed using Image Lab (Bio-Rad). REST mAb (12C11-1) was kindly provided by D J Anderson (*Chen et al., 1998*).

## Image acquisition

Low power imaging of tissue sections was performed in an Olympus SZX10 fluorescence stereomicroscope. For microscopic imaging, 1024 × 1024 pixel pictures were taken with an Olympus IX81 inverted laser scanning confocal microscope equipped with Fluoview 1000 software.

## Immunocytochemistry

Immunocytochemistry was performed as previously described (*Refojo et al., 2011*). The following antibodies were used: chicken anti-GFP 1:3000 (Abcam, Cambridge, MA) (only when specified. In most cases, native GFP is imaged), rabbit anti-GFAP 1:2000 (DAKO, Carpinteria, CA), mouse anti-NeuN 1:1000 (Millipore), rabbit anti-Iba1 1:1000 (WAKO, Osaka, Japan).

## RNA isolation and qRT-PCRs

Total RNA was prepared from cells using mirVana miRNA Isolation Kit (Life Tecnologies). cDNA was generated using Reverse Trancriptase Superscript II and oligo-dT primers (Invitrogen). Quantitative real-time PCR was performed with a Light Cycler (Roche) using QuantiFast SYBR Green PCR Kit (Qiagen, Hilden, Germany). Primer sequences are listed in *Supplementary file 2*. To quantify miR-9 levels, TaqMan probes and Universal Master Mix II were used (Life Technologies).

## Statistical analysis

Each set of numerical data shown was obtained at least in three independent experiments. Statistical analysis was carried out using GraphPad Prism 5 software (Graph Pad Software, San Diego, CA). All values are given as mean ± standard error of the mean (SEM). Statistical significance was assessed as indicated by Student's t-test, one-way ANOVA or two-way ANOVA with repeated measures followed by Bonferroni's post hoc comparisons when appropriate. Differences were considered statistically significant at $*p < 0.05$ and $**p < 0.01$.

## Acknowledgements

We thank A Moebus, S Bauer, S Weidemann, A Tasdemir, and A Krause for their technical assistance, R Kuhn for help with blastocyst injection, N Prakash for critical reading of this manuscript, and A Chen for general advice on the project. We are grateful to DJ Anderson for sharing anti-REST antibody, DC Lie for providing CAG-mRFP, T Kawauchi for CAG-Map1b, and A Fisher for shRNA-REST plasmids.

## Additional information

### Funding

| Funder | Grant reference number | Author |
| --- | --- | --- |
| European Union | SyBoSS FP7-Health-F4-2010-242129 | Wolfgang Wurst |
| Bavarian State Ministry of Education, Science and Arts | Bavarian Research Network - ForIPs | Wolfgang Wurst |
| Bundesministerium für Bildung und Forschung | FKZ 01GN1009C | Wolfgang Wurst |

| Funder | Grant reference number | Author |
|---|---|---|
| Bundesministerium für Bildung und Forschung | FKZ 01GS08151 | Jan M Deussing |
| Bundesministerium für Bildung und Forschung | FKZ 01GS08155 | Jan M Deussing |
| Max-Planck-Gesellschaft | Institute of Psychiatry | Damian Refojo, Jan M Deussing |

The funders had no role in study design, data collection and interpretation, or the decision to submit the work for publication.

## Author contributions

SAG, Designed, executed and analyzed most of the experiments, Wrote the manuscript; AMV, CAV, Contributed with molecular biology and histological studies; MMB, Conducted and analyzed the data of electrophysiology experiments; MLR, Assisted with the ES cell culture; DT, Performed the in silico analysis; WW, DC, JMD, Contributed in the experimental design, Commented on the manuscript; VS, Conceived and analyzed electrophysiology data, Revised the manuscript; DR, Conceived and supervised the study, Wrote the manuscript

## Ethics

Animal experimentation: Animal experiments were conducted under the regulations and protocols for animal experimentation by the local government authorities (the 'Regierung von Oberbayern' Munich, Germany, Permit No. 55.2-1-54-2532-168-10). Every attempt was made to ensure minimum discomfort to the animals at all times.

# Additional files

### Supplementary files

• Supplementary file 1. Predicted miR-9 targets screened in miR-9-Sp[Nestin-Cre] mice.

• Supplementary file 2. List of primers used in qPCR experiments.

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
