## [Decision Letter]

Thank you for sending your work entitled “MicroRNA-9 controls dendritic development by targeting REST” for consideration at *eLife.* Your article has been favorably evaluated by a Senior editor and 3 reviewers, one of whom, Sacha Nelson, is a member of our Board of Reviewing Editors, and one of whom, Jenny Hsieh, has agreed to reveal her identity.

The Reviewing editor and the other reviewers discussed their comments before we reached this decision, and the Reviewing editor has assembled the following comments to help you prepare a revised submission.

All three reviewers felt that this was a detailed and carefully carried out study that revealed a novel function for a broadly expressed microRNA in the regulation of dendritic arborization via regulation of the repressor REST. Although the reviewers felt that the available evidence mostly supported the conclusions reached there were several issues raised that require revision, and at least two of these will require additional experimental work.

All three reviewers wondered about how general the effects were throughout the nervous system. One reviewer noted that prior published reports indicate that REST expression persists in some neuronal populations in hippocampus and neocortex and perhaps elsewhere and so suggest that the authors should consider the possibility that the role of miR-9/REST in dendritic growth regulation may differ as a function of development and/or which neuronal populations are assayed. Given the broad expression of miR-9, the other reviewers wondered about phenotypes at the level of the whole animal resulting from widespread changes in dendritic and axonal arborization. Although a more detailed functional analysis would improve the paper, one relatively simple suggestion is to determine more carefully whether or not there is any gross anatomical change in brain size or cortical thickness, or a consistent change in the soma size of affected hippocampal neurons. These differences can be readily seen in some developmental disorders that produce more modest changes in dendritic arborization than those seen here.

A second issue concerns a missing negative control. The conclusion that REST is the only relevant target for the phenotype would be more convincing if the authors demonstrated that other targets (e.g. CREB or Map1b) are not implicated in miR-9 dependent dendritic branching through a similar series of experiment as was performed for REST. The Map1b target is particularly interesting given its involvement in neurite outgrowth in general, and a recent report demonstrating miR-9 regulation of axonal length via Map1b regulation (Dajas-Bailador). Manipulation of Map1b expression could serve as an important control ideally by recapitulating the previously reported Map1b-dependent axonal phenotype in the sponge mice in the absence of a Map1b-dependent dendritic phenotype. Less ideal, but requiring less additional work, would be to show the lack of a Map1b-dependent dendritic phenotype and acknowledge in the discussion the prior publication and the possibility that other miR-9 targets contribute to axonal phenotypes which were not directly measured.

Minor comments:

1) The morphological analyses of dendrites are relatively complete, but the physiological analysis is somewhat superficial. The authors should probably hedge their bets by noting that the reduced field potential amplitude could be influenced by other factors not measured including neuronal excitability and levels of inhibition.

2) In Figure 4 compared to Figure 4 and C-both the dendritic length and Sholl analysis are on vastly different scales. Why?

---

## [Author Response]

*All three reviewers felt that this was a detailed and carefully carried out study that revealed a novel function for a broadly expressed microRNA in the regulation of dendritic arborization via regulation of the repressor REST. Although the reviewers felt that the available evidence mostly supported the conclusions reached there were several issues raised that require revision, and at least two of these will require additional experimental work*.

*All three reviewers wondered about how general the effects were throughout the nervous system. One reviewer noted that prior published reports indicate that REST expression persists in some neuronal populations in hippocampus and neocortex and perhaps elsewhere and so suggest that the authors should consider the possibility that the role of miR-9/REST in dendritic growth regulation may differ as a function of development and/or which neuronal populations are assayed. Given the broad expression of miR-9, the other reviewers wondered about phenotypes at the level of the whole animal resulting from widespread changes in dendritic and axonal arborization. Although a more detailed functional analysis would improve the paper, one relatively simple suggestion is to determine more carefully whether or not there is any gross anatomical change in brain size or cortical thickness, or a consistent change in the soma size of affected hippocampal neurons. These differences can be readily seen in some developmental disorders that produce more modest changes in dendritic arborization than those seen here*.

We thank the first reviewer for pointing out this very thoughtful comment. We have acknowledged the fact that REST levels are restored in some neuronal populations following a biphasic expression (28; 48), suggesting that miR-9/REST regulation might play other non-developmental roles in some mature neuronal populations This is included in the Discussion section.

Regarding the phenotypes at the level of the whole animal resulting from widespread changes in dendritic and axonal arborization, we have observed, among other behavioral tasks designed to assess sensorimotor performance, that the acoustic startle response is strongly impaired in miR-9-Sp^Nestin-Cre^ mice. This highlights the functional impact from the morphological and electrophysiological impairments. However these studies are still preliminary and other batteries of behavioral tests to assess cognition, emotional-like states and locomotion are still ongoing. The completion of this comprehensive behavioral characterization will take few additional months and consequently we believe that those results will be more suitable for a further study.

We have now extended our morphological analysis finding that the dendritic defects in miR-9 sponge expressing neurons are not restricted to hippocampal neurons but are also observed in cortical neurons (layer V), suggesting that this is a generalized effect of miR-9 loss of function. These results are now included in Figure 4—figure supplement 2. Accordingly we describe this new set of experiments in the Results section. Despite this fact, miR-9-Sp^Nestin-Cre^ mice did not present any gross anatomical change in brain size and also no significant differences were found in cortical thickness measured both directly on tissue sections or using NMR (e.g. data of brain sections: miR-9-Sp^fl-Stop^: 1061 ± 18.60 μm vs. miR-9-Sp^Nestin-Cre^: 1034 ± 22.20 μm, n.s, Student’s *t* test).

It is important to note at this point that alterations in dendritic complexity do not always correlate with brain or cortical size changes. Several mouse models showing diminished dendritic arborization do not have reduced brain size (Krey et al., 2013; Hoogenraad et al., 2005; Henkemeyer et al., 2003). Similarly, dendritic overgrowth can occur without changes in brain size (Jiang et al., 2013; Collins et al., 2004). This suggests that other factors, not directly assessed in this study, might compensate variations in dendritic arborization preserving brain size.

*A second issue concerns a missing negative control. The conclusion that REST is the only relevant target for the phenotype would be more convincing if the authors demonstrated that other targets (e.g. CREB or Map1b) are not implicated in miR-9 dependent dendritic branching through a similar series of experiment as was performed for REST. The Map1b target is particularly interesting given its involvement in neurite outgrowth in general, and a recent report demonstrating miR-9 regulation of axonal length via Map1b regulation (Dajas-Bailador). Manipulation of Map1b expression could serve as an important control ideally by recapitulating the previously reported Map1b-dependent axonal phenotype in the sponge mice in the absence of a Map1b-dependent dendritic phenotype. Less ideal, but requiring less additional work, would be to show the lack of a Map1b-dependent dendritic phenotype and acknowledge in the discussion the prior publication and the possibility that other miR-9 targets contribute to axonal phenotypes which were not directly measured*.

The reviewers address here an important point and we have followed this suggestion.

First we would like to underline that we have not claimed that REST is the only relevant target of miR-9, but on the contrary we consider that it is unrealistic to envision that all the effects observed in miR-9-Sp^Nestin-Cre^ mice might be explained by the regulation of only one single target. Concretely, in the reversion experiments depicted in Figure 5, we show that the expression of shRNAs against REST rescue the effects triggered by the sponge. However the reversion did not reach the 100%, suggesting that other targets beyond REST might also contribute, though in a minor proportion, to the phenotypes observed in miR-9-Sp^Nestin-Cre^ mice. This idea is now expressed in the Discussion.

Regarding the possibility that other miR-9 targets (e.g. Map1b) contribute to axonal phenotypes, we would like to point out that the excellent description of Dajas-Bailador and colleagues of the role of miR-9/Map1b on axonal growth, prompted us to focus in the less explored dendritic role of this microRNA, leaving the well-described axonal phenotypes out of the scope of this study. Following the suggestion of the reviewers, we added an additional layer of complexity to the interpretation of the electrophysiology results. In the Discussion section, we mention that even though impairments in neurotransmission are in agreement with neurons bearing less-arborized dendritic trees, a potential contribution of the tandem miR-9/Map1b, which has already been described to be critical for terminal axonal arborization (20) might also contribute to the functional phenotypes observed.

Beside these considerations, we followed the suggestions of the reviewers and analyzed the effect of shRNAs against Map1b on the dendritic arborization in a similar series of experiments as performed for REST. During the course of the experiments we observed that the shMap1b had a strong effect *per se* on the dendritic development of control neurons. As can be observed in the Figure 6, downregulation of Map1b results in the absence of normal dendrites that are replaced by long filopodia-like structures. Unfortunately, due to this strong *per se* effect, we could not use this tool to evaluate a potential rescue of the dendritic impairment in sponge expressing neurons. Similar effects were observed with a different shRNAi indicating that the results observed are not due to off-target effects of the silencing sequence.Author response image 1.Representative images of primary hippocampal neurons transiently transfected with the indicated constructs and fixed at DIV10. mRFP was cotransfected for visualization. Scale bar 25 μm.

As an alternative, we evaluated the effect of overexpressing the top-three candidates of our previous screening: Foxp4, Creb1 or Map1b on dendritic length and complexity of *wild-type* neurons. Together with REST, those were the highest upregulated targets found in miR-9-Sp^Nestin-Cre^ neurons (Figure 5). We reasoned that if any of those miR-9 targets would contribute to the dendritic phenotype observed in sponge expressing neurons, their overexpression should negatively regulate dendritic growth. In contrast to REST overexpression, we found that increased levels of Foxp4, Creb1 or Map1b had no impact on dendritic length and complexity. These results are now included in Figure 5—figure supplement 1. It is worth noting that even though the down-regulation of Creb1 sensibly impairs activity-dependent dendritic arborization (Wayman et al., 2006), the forced expression of Creb1 does not trigger the opposite effects. Similar to our results obtained in hippocampal neurons Redmond and colleagues (2002) have sown that Creb1 overexpression in culture cortical neurons during development does not exert any effects on dendritic differentiation.

*Minor comments*:

*1) The morphological analyses of dendrites are relatively complete, but the physiological analysis is somewhat superficial. The authors should probably hedge their bets by noting that the reduced field potential amplitude could be influenced by other factors not measured including neuronal excitability and levels of inhibition*.

We extended the electrophysiological characterization of miR-9-Sp^Nestin-Cre^ mice by testing the excitability of CA1 pyramidal cells. Our data show that the action potential (AP) threshold is shifted to more positive membrane potentials in sponge-expressing neurons. The lower excitability leads to fewer AP in neurons projecting onto CA1 pyramidal neurons, thereby strongly reducing the spontaneous EPSC (sEPSC) frequency. In addition, we recorded miniature EPSCs (mEPSC). Here, mEPSC amplitude and frequency were significantly reduced. The reduced frequency reflects the reduced number of synapses, while the reduced mEPSC amplitude indicates a change in quantal size. To exclude presynaptic effects we tested paired-pulse facilitation, a sensitive measure of changes in the probability of transmitter release, which was unchanged. These results are now included in Figure 4 and Figure 4—figure supplement 3 and in the Results section.

*2) In*
Figure 4
*compared to*
Figure 4
*and C-both the dendritic length and Sholl analysis are on vastly different scales*. *Why?*

The reviewers are absolutely right. Two factors might contribute to the differences in dendritic length and complexity between the measurements performed in Thy1-eGFP mice (Figure 4) and the in utero electroporation experiments (Figure 4). First, differences in the genetic backgrounds; Thy1-eGFP mice are in a C57BL/6J background while CD1 mice were used for in utero electroporation. Second, and in our view more relevant, the relative high density of transfected neurons in in utero electroporated brains makes sometimes difficult to discriminate between terminal branches belonging to the neuron being traced or to other neighboring neuron. Since these terminal branches from non-clear sources are excluded from the analyses, tracings from these experiments tend to underestimate the real complexity of the neurons, yielding lower values for total dendritic length and neuronal complexity. Since the same criteria apply to both control- and sponge-transfected groups, this does not represent an intrinsic confounding factor in the comparison between these experimental groups.

References

Collins AL, Levenson JM, Vilaythong AP, Richman R, Armstrong DL, Noebels JL, David SJ, Zoghbi HY (2004) Mild overexpression of MeCP2 causes a progressive neurological disorder in mice. Hum Mol Genet 13:2679-2689.

Dajas-Bailador F, Bonev B, Garcez P, Stanley P, Guillemot F, Papalopulu N (2012) microRNA-9 regulates axon extension and branching by targeting Map1b in mouse cortical neurons. Nat Neurosci 15:697-699.

Gao Z, Ure K, Ding P, Nashaat M, Yuan L, Ma J, Hammer RE, Hsieh J (2011) The master negative regulator REST/NRSF controls adult neurogenesis by restraining the neurogenic program in quiescent stem cells. J Neurosci 31:9772-9786.

Henkemeyer M, Itkis OS, Ngo M, Hickmott PW, Ethell IM (2003) Multiple EphB receptor tyrosine kinases shape dendritic spines in the hippocampus. J Cell Biol 163:1313-1326.

Hoogenraad CC, Milstein AD, Ethell IM, Henkemeyer M, Sheng M (2005) GRIP1 controls dendrite morphogenesis by regulating EphB receptor trafficking. Nat Neurosci 8:906-915.

Jiang M, Ash RT, Baker SA, Suter B, Ferguson A, Park J, Rudy J, Torsky SP, Chao HT, Zoghbi HY, Smirnakis SM (2013) Dendritic arborization and spine dynamics are abnormal in the mouse model of MECP2 duplication syndrome. J Neurosci 33:19518-19533.

Krey JF, Pasca SP, Shcheglovitov A, Yazawa M, Schwemberger R, Rasmusson R, Dolmetsch RE (2013) Timothy syndrome is associated with activity-dependent dendritic retraction in rodent and human neurons. Nat Neurosci 16:201-209.

Kuwabara T, Hsieh J, Nakashima K, Taira K, Gage FH (2004) A small modulatory dsRNA specifies the fate of adult neural stem cells. Cell 116:779-793.

Redmond L, Kashani AH, Ghosh A (2002) Calcium regulation of dendritic growth via CaM kinase IV and CREB-mediated transcription. Neuron 34:999-1010.

Wayman GA, Impey S, Marks D, Saneyoshi T, Grant WF, Derkach V, Soderling TR (2006) Activity-dependent dendritic arborization mediated by CaM-kinase I activation and enhanced CREB-dependent transcription of Wnt-2. Neuron 50:897-909.